# Association Between Peach and Olive Pollen Non-Specific Lipid Transfer Protein Allergy and HLA Class II Phenotype

**DOI:** 10.3390/ijms26167755

**Published:** 2025-08-11

**Authors:** Paula Álvarez, Juan Molina, Raquel Bernardo, Rafael González, Bárbara Manzanares, Rocío Aguado, Laura Carrero, Aurora Jurado, Berta Ruiz-León, Ana Navas

**Affiliations:** 1Department of Immunology and Allergy, Reina Sofía University Hospital, 14004 Córdoba, Spain; z92alrop@uco.es (P.Á.); jeduar.molina.sspa@juntadeandalucia.es (J.M.); raquel.bernardo.sspa@juntadeandalucia.es (R.B.); rafael.gonzalez.sspa@juntadeandalucia.es (R.G.); barbara.manzanares.sspa@juntadeandalucia.es (B.M.); rocio.aguado.sspa@juntadeandalucia.es (R.A.); lauraf.carrero.sspa@juntadeandalucia.es (L.C.); mb.ruiz.sspa@juntadeandalucia.es (B.R.-L.); amaria.navas.sspa@juntadeandalucia.es (A.N.); 2Maimonides Biomedical Research Institute of Córdoba (IMIBIC)/Reina Sofía University Hospital, University of Córdoba, 14004 Córdoba, Spain; 3Allergy Network ARADyAL, Instituto de Salud Carlos III, 28029 Madrid, Spain

**Keywords:** HLA antigens, T-cell epitopes, non-specific lipid transfer proteins, Pru p 3 allergen, Ole e 7 allergen

## Abstract

Concomitant sensitisation to non-specific lipid transfer proteins (nsLTPs) from olive pollen (Ole e 7) and peach (Pru p 3) has been observed in the south of Spain. In the search for reasons to explain this observation, we studied a potential causal relationship between Human Leukocyte Antigen (HLA) molecules and nsLTP sensitisation. For this purpose, eighteen Ole e 7-monosensitised (MONOLE) patients, 22 Pru p 3-monosensitised (MONPRU) patients, and 22 bisensitised (BI) patients were genotyped for HLA class II alleles. Complementarily, T-cell epitopes were predicted with the Immune Epitope Database analysis tool to test HLA epitope presentation. Our results showed a significant increase in DRB1*11 and DQB1*03 frequencies in MONPRU patients and DRB1*04 frequency in MONOLE patients. Additionally, T-cell epitope analysis revealed high binding affinity between the predicted Pru p 3 epitopes and DRB1*11 and between the predicted Ole e 7 epitopes and DRB1*04, suggesting that presentation of these epitopes may be favoured and predisposing individuals to sensitisation. Conversely, low DQB1*05 frequency and poor binding ability of predicted epitopes from both nsLTPs postulated this allele as a possible protective factor to sensitisation. Variations in the binding affinity between nsLTP epitopes and HLA molecules may underlie individual susceptibility to nsLTP allergy.

## 1. Introduction

Non-specific lipid transfer proteins (nsLTPs) are a family of proteins widely distributed among plant kingdom. They are small proteins, ranging from 9 to 11 kDa, that adopt a tunnel-like structure formed by four alpha helices. This structure contains an internal hydrophobic cavity which enables the binding of lipid ligands, the reason why they receive their name [1,2,3]. The most notable feature of nsLTPs is their remarkable structural stability, conferred by four disulphide bridges between eight highly conserved cysteine residues. This structural motif has been identified even in phylogenetically distant species [4,5,6,7,8]. As a result, nsLTPs retain their conformation after exposure to protease digestion and heat treatments [9].

The prevalence of allergy to nsLTP has been extensively documented across Europe, particularly in Mediterranean countries, where these proteins are the leading cause of food allergies in adults [10,11]. One of the main nsLTPs involved is Pru p 3, the nsLTP from peach (*Prunus persica*), which can induce not only oral but also cutaneous and even systemic symptoms [12,13,14]. Pru p 3 allergy is usually accompanied by allergy to other nsLTPs, especially to food, a multi-sensitisation condition known as LTP syndrome [3,13], due to the presence of aminoacid sequence homology between them. Furthermore, olive pollen (*Olea europaea*) allergy is highly prevalent in the southernmost regions of Europe [15,16]. Its nsLTP, Ole e 7, although considered a minor allergen, is responsible for numerous respiratory symptoms of varying severity, particularly in areas with elevated environmental pollen pressure [7]. The presumably high prevalence of co-sensitisation to both nsLTPs observed in some areas of Andalusia (Spain), where olive trees are extensively cultivated, has focused our attention. Previous studies have suggested a possible cross-reactivity between Ole e 7 and Pru p 3 [17,18], although other factors could be influencing the concomitant sensitisation observed.

Since the hygiene hypothesis was first proposed by epidemiologist David Strachan in 1989, knowledge about allergy-triggering determinants has evolved. Today, environmental and genetic factors, along with environmentally induced epigenetic changes, are recognised to contribute to allergic diseases [19]. Among the most extensively studied genes due to their association with immune-mediated disorders, including allergies, are those encoded in the Major Histocompatibility Complex (MHC) locus on the short arm of chromosome 6. The proteins encoded by these genes are collectively referred to as Human Leukocyte Antigens (HLAs). These genes are highly polymorphic, according to their role as antigen-presenting molecules. Particularly, classical HLA class II proteins, HLA-DPA, DPB, DQA, DQB, DRA, and DRB, have been described to be key players in the sensitisation phase of allergy disease [20].

Professional antigen-presenting cells (APCs) or tissue-specialised cells capture nsLTPs allergens from the nasal or digestive epithelium and occasionally from the skin. Then, they are internalised towards the lysosome route and exposed processed peptides, ranging from 10 to 30 amino acids, in HLA class II molecules. These proteins are composed of two chains, alpha and beta, which conform to the peptide-binding groove. The fragmented protein adopts a linear conformation, and the lateral chain of specific amino acids interacts with pockets within the peptide-binding groove of the HLA molecule. Due to the polymorphism of HLA genes, different alleles could favour the binding of different peptides. Once exposed, the HLA-peptide complex is recognised by specific T-cell receptors (TCR) on CD4+ T-cells, which stimulates specific antibody production by B cells and phagocytosis by macrophages [20]. In the context of a type 2 immune response, characterised by the dominance of interleukin (IL)-4, and IL-13, T naïve cells primed with nsLTP peptides differentiate into a Th2 phenotype [21], amplifying the cytokine feedback and prompting B-cell switching towards allergen-specific IgE production. Therefore, the interaction between the HLA molecule and the allergen is crucial to trigger the allergic response.

To define the role played by allergens in the recognition by T- and B-cell receptors, authors have analysed them in the search for epitopes. Peach nsLTP epitopes for B and T-cells have been previously characterised [18,22,23,24,25]; however, studies on Ole e 7 allergy are scarce, with even fewer focusing on epitope mapping. Our group recently described conformational B-cell epitopes for Ole e 7 [18], although T-cell epitope remains undefined.

Given that the prevalence of nsLTP allergy and the frequency of HLA alleles exhibit distinct geographical distribution and that co-sensitisation to both Ole e 7 and Pru p 3 remains a finding with an incompletely understood underlying cause, it would be of interest to investigate a possible interaction between the most frequent HLA genotypes in individuals sensitised to Ole e 7, Pru p 3, or both proteins in our region.

For this purpose, we analysed the distribution of HLA allele frequencies in a population from southern Spain, exposed to high loads of olive pollen, and diagnosed with allergy to Pru p 3 and/or Ole e 7. In addition, we performed an in silico analysis of the interaction between whole Ole e 7 and Pru p 3 proteins and the most frequent obtained HLA molecules to predict T-cell potential epitopes.

## 2. Results

### 2.1. Clinical and Demographic Variables

Demographic characteristics are collected in Table 1. The mean age of the population is 36.1 ± 11.0 years old. Females represent 67.7% of the study population and are also predomintants in MONOLE and MONPRU groups. A total of 53.2% of participants live in rural areas, either surrounded by or closed to olive tree crops, homogeneously distributed across the groups. Regarding symptomatology, rhinoconjunctivitis and asthma are significantly higher in Ole e 7-sensitised patients; however, a reasonably number of MONPRU patients experience respiratory symptoms due to sensitisation to other aeroallergens. The allergic reactions associated with the consumption of *Rosaceae* fruits include local symptoms restricted to the oral cavity, disseminated cutaneous symptoms (pruritus, urticaria, edema), respiratory and/or cardiac involvement, and anaphylaxis. In the MONOLE group, one patient, 5.6% reported oral symptoms related to the ingestion of *Rosaceae* fruits, whereas 31.8% of bisensitised described them. In contrast, MONPRU patients primarily reported skin symptoms (59.1%) after contact with or ingestion of fruits of the *Rosaceae* family, significantly more frequent than in BI patients (13.6%). Moderate to severe symptoms were described only in three bisensitised patients. The levels of specific IgE (sIgE) to Pru p 3 and to Ole e 7 were significantly different between groups as expected, since the classification criteria were based on these values. Additionally, IgE against Ole e 1, the major allergen of olive pollen, was also quantified in patients sensitised to Ole e 7 (Table 1).

### 2.2. Prick Test Results

All of the patients in the MONOLE group tested positive to Olea europea extract and negative to Pru p 3 in the skin prick test (Table 1). Among 18 BI patients tested for SPT, 94.4% (17 out of 18) showed a positive skin prick test to olive extract, and 77.8% (14 out of 18) also tested positive to Pru p 3. Among MONPRU group, 95% (19 out of 20 patients tested) resulted positive to Pru p 3 in the skin prick test. Consistent with the clinical characteristics observed in this group, 47.1% of patients (8 out of 17) showed a positive skin prick test to Olea europaea extract despite not being sensitised to Ole e 7.

### 2.3. nsLTP sIgE Profile

Specific IgE to other nsLTPs (Mal d 3, Art v 3, Tri a 4, Jug r 3, Cor a 8, Ara h 9, and Par j 2) was assessed in 15 patients from each sensitisation group (Figure 1). All MONPRU and BI patients tested showed one or more positive results, with Mal d 3 and Ara h 9 being the most prevalent (Figure 1A,B). The sensitisation heatmaps were strikingly similar in these two groups. By contrast, only two MONOLE patient tested positive for Ara h 9 and one borderline positive for Mal d 3, showing markedly lower levels against these and a few other nsLTPs (Figure 1C).

### 2.4. HLA Genotyping

The allele frequencies were compared between the three sensitisation groups and the control group. Comparisons between a unified “Allergy” group and the control cohort were also conducted. High-resolution genotyping was performed in all patients, although allele frequencies at four-digit resolution were detailed only for comparisons showing significant differences.

DRB1*04 allele was significantly more frequent in the allergy group than in controls (21.77% vs. 10.58%, *p* < 0.001) due mainly to the high frequency in MONOLE patients (Table 2, Figure 2). High-resolution genotyping for DRB1*04 revealed DRB1*04:04 was the most prevalent allele in the allergy group. Among the allergy cohort, this allele was more expressed in MONOLE patients compared with MONPRU and BI (22.22% vs. 0% vs. 4.55%, *p* < 0.001) (Table 2).

Furthermore, DRB1*11 was also heterogeneously distributed among sensitisation groups, being notably more prevalent in MONPRU patients (MONOLE 5.56%, MONPRU 36.36%, BI 9.09%, *p* < 0.001) (Table 2). This was also manifested in the pairwise comparisons between MONPRU and MONOLE and between MONPRU and BI (Figure 2). Observed differences in DRB1*11 allele frequency were attributable in particular to DRB1*11:01. Moreover, this allele was significantly more expressed in MONPRU compared with controls (25.0% vs. 4.56%, *p* < 0.001) (Table 2). In the bisensitised patient group, the frequency of DRB1*15 expression was higher than in the control group; however, this difference did not reach statistical significance.

Regarding HLA-DQB1 alleles, the analysis revealed significant differences in the frequency of the DQB1*03 allele between each sensitisation group (MONPRU 30.56%, MONOLE 52.27%, BI 31.82%, *p* = 0.048), predominantly expressed in MONPRU patients (Table 3, Figure 2). Analysis of DQB1*03 alleles at high resolution showed DQB1*03:01 (DQ7) was significantly more prevalent in MONPRU patients compared with MONOLE and BI patients (40.91% vs. 5.55% vs. 9.09, *p* < 0.001) and compared with controls (40.91% vs. 15.89%, *p* < 0.001). DQB1*03:01 frequency in MONPRU patients was even higher that could be expected by linkage disequilibrium with DRB1*04. Additionally, the frequency of HLA-DQB1*03:02 (DQ8) was higher in MONOLE than in the control group (22.22 vs. 5.30, *p* = 0.001) (Table 3). This is consistent with DRB1*04:04-DQB1*03:02 linkage disequilibrium. Although no significant differences were found in bisensitised patients, DRB1*15-DQB1*06 haplotype is more prevalent in the BI group, supported by the higher frequency of DR*15 and DQ*06 in the BI group than in controls.

Conversely, in the analysis of allele frequencies, HLA-DQB1*05 was found to be underrepresented in the allergy group compared to the control population (8.06% vs. 19.87%, *p* = 0.038), in particular in MONPRU patients (Table 3, Figure 2). Differences in the other alleles studied did not reach the level of significance between the allergic groups.

Our analysis showed allergic patients are more likely to be DR4-DQ7, while non-allergic controls are DR7-DQ2, which are alleles usually described in our geographical area. Based on the data shown, DRB1*04 and DQB1*03 (03:02) were more prevalent in patients sensitised to Ole e 7 (MONOLE and BI) than in the general population, while MONPRU patients preferentially expressed DRB1*11 and DQB1*03 (03:01) alleles. The frequency of HLA-DRB1*15 was elevated in the bisensitised patient group; however, the differences compared to the control group and the other two patient groups did not reach statistical significance.

### 2.5. Predicted T-Cell Epitopes for Ole e 7 and Pru p 3

After evaluating the results on allele frequencies in all sensitisation groups, we considered whether antigen presentation might explain the observed differences. Therefore, we next examined the interaction of these HLA molecules with peptides derived from peach and olive pollen nsLTPs using an epitope prediction tool. High-resolution typing was considered in epitope prediction, as differences in a single amino acid can be relevant. The epitope prediction shown in Table 4 corresponds to those alleles whose frequencies were significantly different in the previous analysis and which had a population frequency greater than 2% (Table 2 and Table 3). Among all possible combinations between these selected alleles and possible epitopes, only the one with the highest score was selected.

Henceforth, predicted epitopes will be referred to by the name of the protein (Ole or Pru), followed by the positions of the first and last residues of the epitope, indicated in subscript. The prediction of epitopes derived from Ole e 7 and presented by HLA-DR showed that the HLA-DRB1*04:05 allele exhibited the highest binding affinity for the predicted epitope Ole _54–68_ “KSALALVGNKVDTGR”. The epitope may be considered a strong binder giving its binding score a value of 0.79 and a percentile rank 0.83. This result was followed by other combinations of DRB1*04 alleles (*04:03, *04:04) and peptides, which also exhibited strong binding scores, with DRB1*04:03 sharing the same epitope sequence (Table 4). Conversely, predictions for Ole e 7 epitopes presented by DRB1*11 alleles yielded lower binding scores, even when considering the top-ranking results (Table 4). Regarding HLA-DQ, the predicted T-cell epitopes from Ole e 7 demonstrated high-affinity binding to the DQB1*03:01 allele. The epitope Ole _61–75_ “GNKVDTGRVSSLPKK” from Ole e 7 matched with this allele achieved a score of 0.63 and a percentile rank 1.20.

The alignment of Pru p 3 with HLA-DR revealed that overall, binding scores were higher for predicted epitopes bound to DRB1*11 alleles than for DRB1*04. The DRB1*11:04 allele was able to effectively present the predicted epitope Pru _31–45_ “IRNVNNLARTTPDRQ” receiving a binding score of 0.61. Similarly, the DRB1*11:02 allele presented the epitope Pru _29–43_, “NGIRNVNNLARTTPD,” located near the former epitope, with a binding score of 0.59 (Table 4). The prediction of Pru p 3 peptides in association with HLA-DQ demonstrated high-affinity binding to the DQB1*03:01 allele. The matched epitope Pru _51–65_ “LKQLSASVPGVNPNN” received a score of 0.59 and a percentile rank of 1.5, which may be considered a strong binder. DQB1*05:01 and the predicted epitope obtained very low binding scores, as in the case of Ole e 7-derived epitope.

The prediction of Ole e 7 T-cell epitopes suggested that HLA-DRB1*04 may be a candidate allele for strong peptide presentation from this protein, potentially triggering an immunogenic response, which is consistent with the high prevalence of DR4 in MONOLE and BI patients. Similarly, HLA-DQB1*03 alleles could properly present Ole e 7, hence patients carrying DR4-DQ7 haplotype would be good presenters of Ole e 7-derived peptides. For the prediction of Pru p 3 epitopes, the best parameters were obtained with DRB1*11 and DQB1*03 alleles, which illustrated the high prevalence of DR11-DQ7 haplotype in MONPRU patients. The low frequency of HLA-DQB1*05 among allergic subjects may be attributable to the weak presentation of nsLTP-derived epitopes to T cells.

### 2.6. Correlation Between Specific IgE Levels and HLA Genotyping

Finally, we sought to determine whether sIgE levels were related to the favourable presentation of the proteins, particularly in patients carrying DRB1*04 (DR4) or DRB1*11 (DR11) alleles, both in the overall population and within each sensitisation group. We first presented sIgE levels to Ole e 7 and to Pru p 3 across all sensitisation groups (Table 1, Figure 3A). Although the positivity threshold recommended by the manufacturer to consider a patient sensitised to an allergen was 0.35 kUA/L, most of the samples tested had sIgE levels to Ole e 7 and to Pru p 3 considerably higher than the mentioned threshold.

Regarding mean sIgE levels, Ole e 7 levels were significantly higher than those of Pru p 3 in bisensitised patients (95.30 ± 106.42 kUA/L vs. 16.06 ± 14.06 kUA/L, *p* = 0.002). Analysis by allele revealed that Pru p 3-specific IgE levels were significantly higher in DR11-positive (DR11+) patients compared to DR11-negative (DR11−) patients, both in the overall allergic population (16.96 ± 15.24 vs. 8.36 ± 11.29, *p* = 0.006) and among bisensitised patients (31.74 ± 17.32 vs. 12.57 ± 10.97, *p* = 0.027). Similarly, although not statistically significant, Ole e 7 sIgE levels were higher in patients carrying the DR4 allele (DR4+) compared to those who did not carry this allele (DR4−) within the MONOLE group (64.14 ± 45.94 vs. 49.56 ± 31.71; *p* = 0.5) (Figure 3B).

## 3. Discussion

In the present study, we analysed the role of HLA polymorphisms in the presentation of two allergenic proteins, Ole e 7 and Pru p 3, both belonging to the plant nsLTP family. HLA typing revealed genetically distinct patient groups. In MONOLE patients, the DRB1*04 allele was markedly overrepresented. In contrast, MONPRU patients showed a significant association with DRB1*11, which was even higher for DQB1*03:01. Moreover, the epitope prediction suggests a potential predisposition to sensitisation and allergic response upon exposure to these proteins in (i) patients carrying HLA-DRB1*11 or HLA-DQB1*03:01, which strongly presents Pru p 3 peptides; and (ii) patients carrying HLA-DRB1*04, which binds Ole e 7 peptides. Conversely, the absence of patients carrying the HLA-DQB1*05 allele, together with the low binding scores for Ole e 7 and Pru p 3 epitopes associated with this allele, suggests a potential protective role against the development of nsLTP allergy.

Polymorphisms in HLA may influence both antigen binding and the recognition of the antigen-HLA complex by the T-cell receptor. Consequently, the impact of this interaction on the development of the immune response, particularly in relation to susceptibility to certain diseases, has been extensively studied. Diseases with the strongest HLA associations include coeliac disease (approximately 90% of affected individuals carry the HLA-DQ2.5 heterodimer, and to a lesser extent HLA-DQ8, DQ2.2 or DQ7.5) [26], ankylosing spondylitis (approximately 90% are HLA-B27 positive) [27], birdshot chorioretinopathy (95% are HLA-A29 positive) [28], and type 2 autoimmune hepatitis (93% are HLA-DQ2 positive) [29]. Numerous associations have been reported between HLA genes—both class I and class II—as well as non-classical HLA genes, such as HLA-G, with considerable variation depending on the study population. Thus, a particular allele may confer susceptibility to a specific allergen in a given population, whereas in a different population or in the context of a different allergen, the same allele may not confer susceptibility or may even be associated with a protective effect [20].

Other authors have further investigated HLA associations not only with disease, but also with allergic responses to pollen, mites, venoms, or drugs [30,31,32,33,34], as well as to food allergy. A genome-wide association study (GWAS) conducted in a Japanese population revealed a strong association between peach allergy and HLA genes, specifically the HLA alleles DRB1*09:01 and DQB1*03:03 [35]. In 2018, Nucera et al. published a report on HLA-DRB1 genotyping in Italian patients allergic to nsLTPs [36]. The study included a range of nsLTP sources, such as hazelnut, chestnut, peach, kiwi, peanut, and cherry, and compared these patients to individuals allergic to non-LTP allergens and to a healthy control population. The researchers found that the frequencies of the HLA-DRB1*07 and HLA-DRB1*14 alleles were significantly lower in nsLTP-positive patients compared to nsLTP-negative individuals. In our study, these alleles were also found to be less frequent in allergic subjects compared to the control population (Table 2). However, as the allergen sources are not identical, these data may not be directly comparable. In the context of HLA allele associations with food allergens, Nucera et al. reported a negative association between the peach nsLTP allergen and the HLA-DRB1*04 allele [36]. In our cohort, MONPRU patients exhibited a lower prevalence of HLA-DRB1*04 compared to the overall allergic population (13.64% vs. 21.77%, Table 2). Additionally, these authors described an association between peach allergy and the DRB1*11/13 haplotype, which aligns with our finding of a strong association between peach allergy and DRB1*11. Although both the allergic and control populations in the study by Nucera et al. were from Italy, the prevalence of nsLTP allergy in Italy and Spain is relatively comparable due to their shared location within the Mediterranean basin. In our study, the control population was specifically selected to minimise geographical bias, as regional differences in the prevalence of certain alleles have been reported even within different regions of Spain [37].

Several studies have previously described Pru p 3 epitopes using different approaches. In 2009, Tordesillas et al. identified two main T-cell epitopes of Pru p 3 using randomly generated decamer (10-mers) peptides [24]. The Pru p 3 epitopes predicted in our study for antigen presentation by DRB1*11:02 and DRB1*11:04, Pru _29–43_ and Pru _31–45_ (Table 4), share more than half of their amino acid sequence with one of the epitopes described by Tordesillas et al. Furthermore, in the same year, Schulten et al. generated Pru p 3-specific T-cell clones from samples obtained from Pru p 3-allergic patients in Italy and Spain [38]. Using this approach, they identified four regions most frequently recognised by the T-cell clones: Pru p 3 (13–27), Pru p 3 (34–48), Pru p 3 (43–57), and Pru p 3 (61–75). Additionally, Pastorello et al. described Pru p 3 T-cell epitopes using T-cell lines, assessing their proliferation in response to eight overlapping peptides [25]. Two immunodominant Pru p 3 peptides were identified: Pru p 3_12–27_ and Pru p 3_57–72_. The former matched the peptide predicted in our study to be presented by DQB1*03:01. Pastorello et al. also reported a high frequency of HLA-DRB1*11; however, the peptides we identified as being presented by this allele were not recognised by the majority of T-cell lines.

On the other hand, our group recently published an article on the epitope mapping of these two proteins using liquid chromatography coupled to mass spectrometry (LC-MS) in patients with similar sensitisation profiles. In that study, we were the first to describe two previously unreported B-cell epitopes of Ole e 7 [18]. Despite the limited literature available for comparison, it is noteworthy that the most favourable T-cell epitope predicted in the present study (Table 4) shares its sequence with one of the Ole e 7 epitopes previously identified by our group. The epitopes in question are “KSALALVGNKV” identified through LC-MS mapping and the predicted epitope Ole _54–68_ “KSALALVGNKVDTGR”. In addition, the predicted peptide Ole _14–28_ “CVSYLDDKSAKPTSD” is located within the same region as the other Ole e 7 B-cell epitope described in the aforementioned publication, “KLTSCVSYLDDKS”. These data further support the predicted epitopes.

The aforementioned studies showed different in vitro experiments that could have been performed to validate the results obtained from epitope prediction, such as the culture of peripheral blood mononuclear cells or T-cells with peptides and the subsequent quantification of released cytokines, the measurement of proliferation under peptide stimulation conditions or cellular phenotyping of activated cell populations after incubation [24,25,38]. LC-MS was also used in a previous study for B-cell epitope mapping [18]; however, functional assays are missing in this work, and validation of the prediction tool for the study of epitopes is needed. Therefore, this result should be considered as an additional support for existing evidence on allele frequencies.

Due to the lack of scientific literature on the association between HLA antigens and allergic sensitisation to Ole e 7, we considered studies on other related allergens. Cárdaba et al. investigated the prevalence of HLA alleles in patients allergic to Ole e 1, the major allergen in olive pollen, and reported an association with HLA-DR7 and HLA-DQ2 [39,40]. In our study, we did not observe a predominance of either the DR7 or DQ2 allele in the MONOLE group, nor were there any significant differences in their prevalence compared to the control group. Several years later, in 2021, Gheerbrant et al. analysed the association between respiratory allergy, including Ole e 1 as an allergen, and HLA class II typing in a French population. Among the significant differences identified in patients sensitised to this allergen, the alleles DQA1*02:01, DQB1*02:01, DQB1*03:01, and DRB1*03:01 were reported [41]. Regarding DQ2, Gheerbrant’s findings were consistent with the earlier results reported by Cárdaba et al. [39]. However, in the present study, only DQB1*03:01 was identified as a good presenter of Ole e 7 peptides, although its prevalence in the MONOLE group was similar to that in the control group (Table 3). The distinct differences in the most prevalent alleles between Ole e 1 and Ole e 7 allergic subjects suggest that our results may be attributable to Ole e 7 itself rather than to Ole e 1. Indeed, sIgE to Ole e 1 was measured in MONOLE and BI patients. Among these, four MONOLE and three BI patients showed elevated sIgE levels to Ole e 1 (>20 kUA/L). As predicted by the Cárdaba et al. study [39,40], six of these patients were positive for HLA-DRB1*07, and five of them were also positive for HLA-DQB1*02. These findings support the hypothesis that antigenic presentation for Ole e 1 and Ole e 7 sensitisation is mediated by different HLA molecules, highlighting the role of Ole e 7 as a major allergen in regions with high olive pollen pressure.

One of the main limitations of this study is its geographical setting. On the one hand, the heavy pollen burden in the area leads to a high prevalence of allergy to Ole e 7 which facilitated the recruitment of a substantial group of patients monosensitised to this nsLTP. On the other hand, the high prevalence of olive tree allergy limited the identification of patients monosensitised to Pru p 3 and not sensitised to Ole e 7, as many of them also exhibited seasonal symptoms, and are in fact sensitised to olea pollen.

Additionally, although a cross-sectional design is appropriate for exploring genetic characteristics such as those examined in this study, a longitudinal design would be more suitable for confirming or ruling-out a potential pollen-food syndrome between olive and peach nsLTPs.

Increasing the sample size of the three groups would help verify the associations between each phenotype and specific HLA molecules.

Finally, although in silico T-cell epitope predictions are appropriate, our findings could be validated through functional approaches such as ELISPOT or assays involving HLA-peptide tetramers.

In conclusion, this study provides the first description of HLA alleles in patients allergic to Ole e 7, representing a significant contribution to the understanding of this specific condition. The map of sensitisation to nsLTPs of the MONPRU group is very similar to that of the BI patients and diametrically different from that of the MONOLE subjects. In a geographical region with a high olive pollen pressure, such as the one in which the study was carried out, the clinical spectrum of MONPRU patients is very similar to that of BI patients, with a high percentage of both groups having a positive prick test, both for olive and peach. Despite these similarities, in the same geographical area with high olive pollen load, only certain patients are sensitised to Ole e 7. The way in which HLA molecules bind to Ole e 7 and Pru p 3 T-cell peptides may explain, at least in part, these differences in sensitisation patterns.

Bioinformatics predictive models have proven highly valuable in peptides in certain pathological conditions. In our study, beyond the initial descriptive analysis, we identified T-cell epitopes efficiently presented by the specific HLA molecules most frequently found in patients sensitised to nsLTPs. This information may help bridge a critical knowledge gap and pave the way for the development of truly specific immunotherapies.

## 4. Materials and Methods

### 4.1. Patients and Controls

A total of 62 patients with olive pollen or peach allergy, diagnosed at the Immunology and Allergy Department of Reina Sofia University Hospital (Córdoba, Spain), were consecutively recruited between 2019 and 2024. Inclusion criteria required all patients to test positive for specific IgE against nsLTPs, Ole e 7, and/or Pru p 3 (sIgE > 0.35 kUA/L), accompanied by oral and/or respiratory allergic symptoms. Patients who underwent specific immunotherapy for the allergens under study within the five years preceding the recruitment were excluded. The geographical origin of the patients was restricted to Andalusia, specifically the provinces of Córdoba and Jaén.

Allergic patients were stratified into three groups according to sIgE levels:(i)Patients monosensitised to Ole e 7 (sIgE to Ole e 7 > 0.35 kUA/L and sIgE to Pru p 3 < 0.35 kUA/L) were grouped as MONOLE (n = 18).(ii)Patients monosensitised to Pru p 3 (IgE to Pru p 3 > 0.35 kUA/L and sIgE to Ole e 7 < 0.35 kUA/L) were grouped as MONPRU (n = 22).(iii)Patients sensitised to both allergens (sIgE to Ole e 7 and Pru p 3 > 0.35 kUA/L) were grouped as BI (n = 22).

The allele frequency data from the control population were obtained from a previous study conducted in our department on 548 healthy volunteers. This population came from the same geographical areas as the patients. Samples were collected and analysed following the same protocol as the study group to minimise potential confounding factors.

Clinical allergy diagnoses were made by physicians based on both clinical and laboratory data. Demographic and clinical information was gathered through patient questionnaires and from anamnesis recorded in their medical histories by healthcare professionals.

This study was approved by the Ethics Committee of the Reina Sofia University Hospital (ref. 4508), and written informed consent was obtained from all participants.

### 4.2. IgE Quantification

Specific IgE (sIgE) levels to Ole e 1, Ole e 7 and Pru p 3 were measured in fresh serum samples using the ImmunoCAP 250 system (Thermo Fisher Scientific™, Uppsala, Sweden) following the manufacturer’s instructions. The established threshold for positivity was 0.35 kUA/L. As part of the diagnostic work-up, sIgE to other nsLTPs, including Mal d 3, Art v 3, Tri a 4, Jug r 3, Cor a 8, Ara h 9, and Par j 2, was also assessed in most cases using the same procedure.

### 4.3. Prick Test

The skin prick test (SPT) was performed according to European guidelines [42], using commercial extracts from *Olea europaea* pollen and peach (ALK Abelló, Hørsholm, Denmark). A positive SPT result was defined as a wheal diameter at least 3 mm larger than that produced by the negative control.

### 4.4. HLA Typing

DNA extraction was performed using EDTA-anticoagulated peripheral blood. High-resolution genotyping of HLA-DRB1 and HLA-DQB1 loci was performed using Luminex^®^ xMAP^®^ technology with the LIFECODES^®^ HLA-SSO Typing Kit (catalogue numbers 628925 and 628930, respectively; Immucor, Inc., Norcross, GA, USA). HLA genotype was analysed using Match It! software (1.2 version). Next-generation sequencing (NGS) was used in cases of ambiguous results. For library preparation, NGSgo^®^-MX6-1 (catalogue number: 7971464; GenDx, Utrecht, The Netherlands) for multiplexed HLA amplification (HLA-A, B, C, DRB1, DQB1 and DPB1 alleles) and NGSgo^®^ Library Full Kit: NGSgo-IndX Plate III & NGSgo-LibrX and GenDx-AMPure XP) (catalogue number: 2842356; GenDx) were used. Sequencing was performed in MiSeq^®^ system (Illumina, Inc., San Diego, CA, USA), using MiSeq^®^ v2 Reagent Kit (catalogue number: 15033624, Illumina) and MiSeq^®^ Reagent Micro Kit v2 (catalogue number: 15036715, Illumina).

### 4.5. T Cell Epitope Prediction

T-cell epitopes were in silico predicted with the Immune Epitope Database (IEDB) analysis resource tool by using the full known sequence of Pru p 3 (UniProt code Q9LED1) and the Ole e 7 sequence obtained from the literature [43] and pairing them with HLA class II alleles. The prediction method used was NetMHCIIpan 4.1. Epitope length was 15 amino acids by default. Predicted epitopes were ranked based on binding affinity scores and percentile rank provided by the resource, which indicates the percentage of randomly sampled peptides with better scores than the given peptide. A high predicted binding affinity was considered with scores greater than 0.5, scores between 0.2 and 0.5 indicated moderated predicted binding affinity, and scores below 0.2 indicated a low binding affinity. Epitopes were defined as strong binders when the percentile rank was 2 or less, weak binders with a percentile between 2 and 10. Percentile ranks above 10 indicated a low probability for the peptide to be an epitope [44,45].

The selection of HLA-DR and HLA-DQ loci for epitope prediction was based on the results of the analysis of frequencies at two-digit resolution, choosing those alleles statistically different between allergic groups and control population with a population frequency greater than 2%. Moreover, of all combinations of epitopes, only the one with the best score was selected.

### 4.6. Statistical Analysis

Clinical and demographic variables were analysed using counts and frequencies. Categorical variables were compared between groups using the Chi square test or Fisher’s exact test, as appropriate, and quantitative variables were analysed using Kruskal–Wallis test. *p*-values less than 0.05 were considered statistically significant. Although HLA typing was conducted at a resolution of four digits or higher, statistical comparisons of allelic frequencies were generally restricted to the two-digit level due to sample size constraints; however, in specific instances, four-digit analysis were performed. Bonferroni correction was applied to the significance of the *p*-values for multiple HLA allele comparisons.

## Figures and Tables

**Figure 1 ijms-26-07755-f001:**
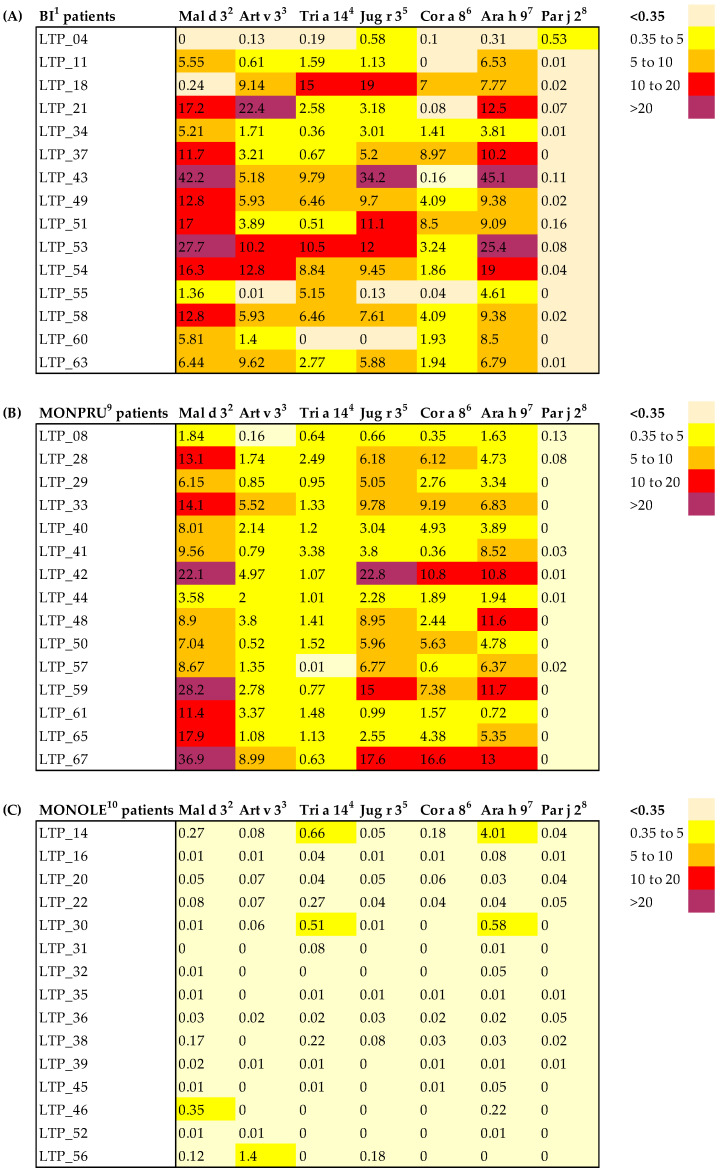
Heatmap of sIgE levels against a selection of nsLTPs. Individual sIgE levels (kUA/L) are shown for each subject, stratified by sensitisation group: BI (**A**), MONPRU (**B**) and MONOLE (**C**) patients. Color intensity represents the degree of sIgE positivity—darker colours indicate greater positivity—and values above 0.35 kUA/L were considered positive. BI^1^: patients bisensitised to Ole e 7 and Pru p 3; Mal d 3^2^: apple nsLTP; Art v 3^3^: mugwort nsLTP; Tri a 14^4^: wheat nsLTP; Jug r 3^5^: walnut nsLTP; Cor a 8^6^: hazelnut nsLTP; Ara h 9^7^: peanut nsLTP; Par j 2^8^: parietaria nsLTP; MONPRU^9^: patients sensitised to Pru p 3; MONOLE^10^: patients sensitised to Ole e 7.

**Figure 2 ijms-26-07755-f002:**
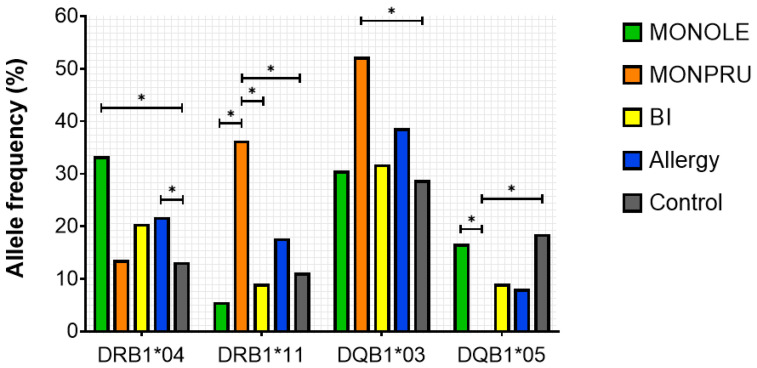
Representation of statistically significant pairwise comparisons of HLA-DRB1 and DQB1 allele frequencies. *: *p*-value < 0.05.

**Figure 3 ijms-26-07755-f003:**
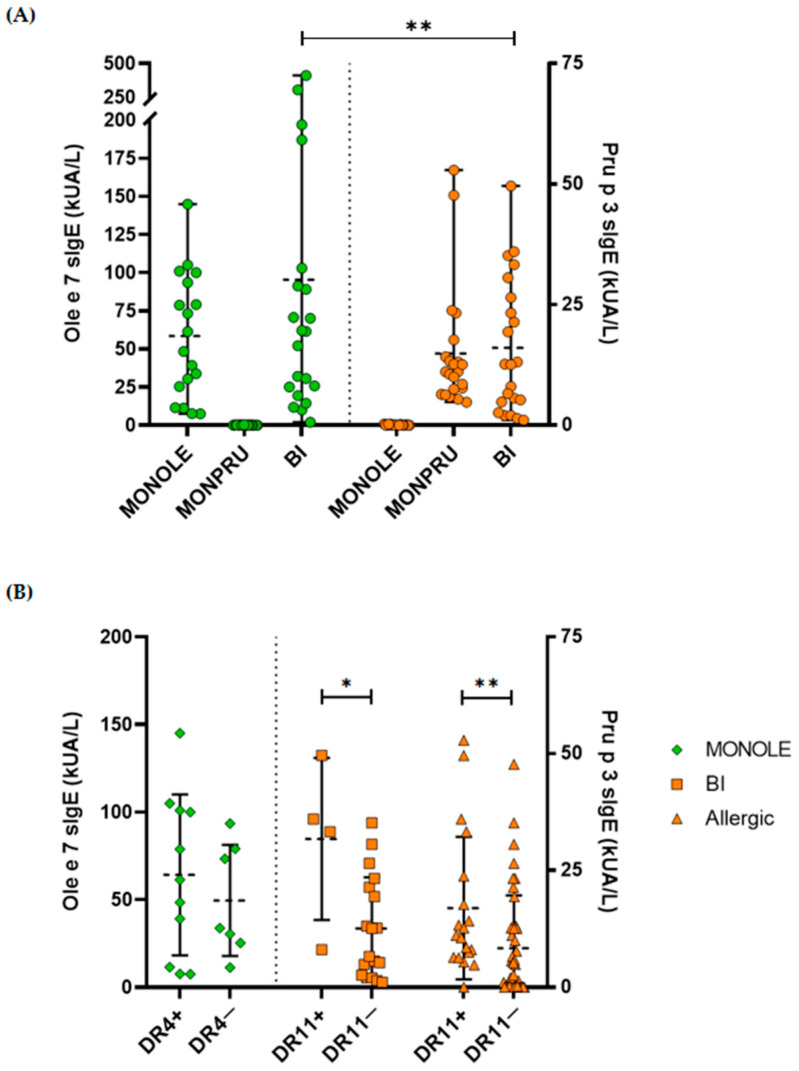
Dot plots for specific IgE (sIgE) circulating levels (kUA/L) to Ole e 7 (green symbols, left axis) and to Pru p 3 (orange symbols, right axis) allergens in the entire allergic cohort according to their sensitisation state (**A**), classified as MONOLE (Ole e 7-monosensitised patients), MONPRU (Pru p 3-monosensitised patients) and BI (Ole e 7 and Pru p 3 bisensitised patients); and according to the presence of DRB1*04 or DRB1*11 HLA-class II alleles genotyped by low resolution (**B**). The sensitisation state of patients was indicated by different symbols: MONOLE patients by diamonds, BI patients by squares and Allergic (MONOLE, MONPRU and BI) patients by triangles. Only comparisons with significant differences are shown. Mean values are displayed by central lines and standard deviation values by whiskers. *: *p*-value < 0.05; **: *p*-value < 0.01.

**Table 1 ijms-26-07755-t001:** Demographic characteristics of the study groups classified by allergic profile.

	MONOLE n = 18	MONPRU n = 22	BI n = 22	*p*-Value ^1^
Age, mean (SD)	38.3 (11.8)	37.0 (10.2)	33.6 (11.1)	0.383
Female, n (%)	14 (77.8)	17 (77.3)	11 (50.0)	0.086
Rural habitat, n (%)	9 (50.0)	13 (59.1)	11 (50.0)	0.790
Respiratory symptoms				
Rhinoconjunctivitis, n (%)	17 (94.4)	15 (68.2)	22 (100.0)	**0.004**
Asthma, n (%)	16 (88.9)	7 (31.8)	15 (68.2)	**0.001**
*Rosaceae* symptoms				
Oral symptoms, n (%)	1 (5.6)	2 (9.1)	7 (31.8)	**0.043**
Mild systemic-cutaneous symptoms, n (%)	0	13 (59.1)	3 (13.6)	**<0.001**
Moderate systemic-respiratory symptoms, n (%)	0	0	2 (9.1)	0.153
Anaphylaxis, n (%)	0	0	1 (4.5)	0.397
sIgE levels				
Pru p 3 sIgE (kUA/L), mean (SD)	0.09 (0.10)	14.85 (12.59)	16.06 (14.06)	**<0.001**
Ole e 7 sIgE (kUA/L), mean (SD)	58.47 (40.62)	0.12 (0.12)	95.30 (106.42)	**<0.001**
Ole e 1 sIgE ^2^ (kUA/L), mean (SD)	17.85 (38.37)	NA	23.73 (44.40)	0.838
Positive SPT ^3^ with Olea europaea extract, n (%)	18 (100.0)	8 (47.1)	17 (94.4)	**<0.001**
Positive SPT ^3^ with peach extract, n (%)	0	19 (95.0)	14 (77.8)	**<0.001**

Symptomatology evaluated respiratory symptoms, including rhinoconjunctivitis and asthma, and symptoms related to the consumption of Rosaceae fruits, stratified by severity. Qualitative variables are described as counts and frequencies (%). Quantification of specific IgE levels is also described. ^1^
*p*-value for the comparison performed between MONOLE, MONOPRU, and BI patients according to Pearson Chi square for qualitative variables and to Kruskal–Wallis for quantitative ones. Significant *p*-values < 0.05 are shown in bold. ^2^ Ole e 1 is measured in 15 out of 18 MONOLE patients and in 16 out of 22 BI patients. Quantification is not applicable (NA) in patients not sensitised to Ole e 7. ^3^ SPT, skin prick test.

**Table 2 ijms-26-07755-t002:** HLA-DRB1 allele frequencies.

	Allergy				Total Allergy Group N = 62	Control Group N = 548	*p*-Value ^2^
	MONOLE N = 18	MONPRU N = 22	BI N = 22	*p*-Value ^1^			
DRB1*01	8.33	0	6.82	ns	4.84	13.68	ns
DRB1*03	13.89	11.36	6.82	ns	10.48	10.95	ns
DRB1*04	33.33	13.64	20.45	ns	21.77	10.58	**<0.001**
DRB1*04:01	0	2.27	0	ns	0.81	1.46	ns
DRB1*04:02	2.78	4.55	2.27	ns	3.23	1.82	ns
DRB1*04:03	5.56	0	4.55	ns	3.23	2.19	ns
DRB1*04:04	22.22	0	4.55	**<0.001**	8.06	2.01	**<0.001**
DRB1*04:05	2.78	4.55	6.82	ns	4.84	2.01	ns
DRB1*04:06	0	0	0	ns	0	0.36	ns
DRB1*04:07	0	2.27	2.27	ns	1.61	0.73	ns
DRB1*07	13.89	9.09	15.91	ns	12.90	16.42	ns
DRB1*08	2.78	4.55	0	ns	2.42	3.10	ns
DRB1*09	0	2.27	4.55	ns	2.42	1.46	ns
DRB1*10	2.78	0	0	ns	0.81	1.46	ns
DRB1*11	5.56	36.36	9.09	**<0.001**	17.74	13.87	ns
DRB1*11:01	2.78	25.0	4.55	**0.001**	11.29	4.56	ns
DRB1*11:02	0	2.27	0	ns	0.81	2.55	ns
DRB1*11:03	0	0	2.27	ns	0.81	1.10	ns
DRB1*11:04	2.78	9.09	2.27	ns	4.84	5.66	ns
DRB1*12	0	0	0	ns	0	0.91	ns
DRB1*13	2.78	11.36	13.64	ns	9.68	12.59	ns
DRB1*14	2.78	0	0	ns	0.81	4.02	ns
DRB1*15	13.89	11.36	20.45	ns	15.32	9.67	ns
DRB1*16	0	0	2.27	ns	0.81	1.28	ns

Data of patients are detailed according to the sensitisation group (MONOLE, MONPRU, BI), as well as combining all allergic subjects into a single “Allergy” group. Allele frequencies (%) were calculated using allele count divided by the total number of alleles (2n) multiplied by 100. High-resolution genotyping is only detailed for those alleles showing significant differences in allele frequency within the allergic cohort or in comparison with the control group. ^1^
*p*-value for the comparison performed between MONOLE, MONOPRU, and BI patients according to Pearson Chi square. ^2^
*p*-value for the comparison performed between allergy and control groups according to Chi square and adjusted using Bonferroni method. Significant *p*-values < 0.05 are shown in bold.

**Table 3 ijms-26-07755-t003:** HLA-DQB1 allele frequencies.

	Allergy				Total Allergy Group N = 62	Control Group N = 151	*p*-Value ^2^
	MONOLE N = 18	MONPRU N = 22	BI N = 22	*p*-Value ^1^			
DQB1*02	30.56	22.73	25.00	ns	25.81	29.14	ns
DQB1*03	30.56	52.27	31.82	**0.048**	38.71	25.83	ns
DQB1*03:01	5.56	40.91	9.09	**<0.001**	19.35	15.89	ns
DQB1*03:02	22.22	9.09	15.91	ns	15.32	5.30	ns
DQB1*03:03	0	2.27	6.82	ns	3.23	3.97	ns
DQB1*03:04	2.78	0	0	ns	0.81	0	ns
DQB1*03:05	0	0	0	ns	0	0.66	ns
DQB1*04	8.33	4.55	0	ns	4.03	1.99	ns
DQB1*05	16.67	0	9.09	**0.038**	8.06	19.87	ns
DQB1*05:01	11.11	0	6.82	ns	5.65	14.57	ns
DQB1*05:02	2.78	0	2.27	ns	1.61	1.99	ns
DQB1*05:03	2.78	0	0	ns	0.81	3.31	ns
DQB1*06	13.89	20.45	34.09	ns	23.39	23.17	ns

Data of patients are detailed according to the sensitisation group (MONOLE, MONPRU, BI), as well as combining all allergic subjects into a single “Allergy” group. Allele frequencies (%) were calculated using allele count divided by the total number of alleles (2n) multiplied by 100. High-resolution genotyping is only detailed for those alleles showing significant differences in allele frequency within the allergic cohort or in comparison with the control group. ^1^
*p*-value for the comparison performed between MONOLE, MONOPRU, and BI patients according to Pearson Chi square. ^2^
*p*-value for the comparison performed between allergy and control groups according to Chi square and adjusted using Bonferroni method. Significant *p*-values < 0.05 are shown in bold.

**Table 4 ijms-26-07755-t004:** Predicted T-cell epitopes for Ole e 7 and Pru p 3 and their binding affinity scores.

Protein	Allele	Peptide	Start Position	End Position	Score ^1^	Percentile Rank ^2^
Ole e 7	DRB1					
	DRB1*04:03	KSALALVGNKVDTGR	54	68	0.73	1.1
	DRB1*04:04	GVKTVLAQATSKPDK	32	46	0.63	2.4
	DRB1*04:05	KSALALVGNKVDTGR	54	68	0.79	0.8
	DRB1*11:01	TGRVSSLPKKCGMSV	66	80	0.33	5.8
	DRB1*11:02	TGRVSSLPKKCGMSV	66	80	0.19	13.0
	DRB1*11:04	TGRVSSLPKKCGMSV	66	80	0.54	3.3
	DQB1					
	DQB1*03:01	GNKVDTGRVSSLPKK	61	75	0.63	1.2
	DQB1*03:02	TAKLTSCVSYLDDKS	8	22	0.01	2.5
	DQB1*03:03	GNKVDTGRVSSLPKK	61	75	0.24	0.69
	DQB1*05:01	TSCVSYLDDKSAKPT	12	26	0.02	2.1
	DQB1*05:03	TSCVSYLDDKSAKPT	12	26	0.06	2.5
Pru p 3	DRB1					
	DRB1*04:03	NVNNLARTTPDRQAA	33	47	0.38	6.5
	DRB1*04:04	NGIRNVNNLARTTPD	29	43	0.32	7.2
	DRB1*04:05	SIPYKISASTNCATV	76	90	0.16	11.0
	DRB1*11:01	IRNVNNLARTTPDRQ	31	45	0.33	5.9
	DRB1*11:02	NGIRNVNNLARTTPD	29	43	0.59	2.5
	DRB1*11:04	IRNVNNLARTTPDRQ	31	45	0.61	2.7
	DQB1					
	DQB1*03:01	LKQLSASVPGVNPNN	51	65	0.59	1.5
	DQB1*03:02	VPGVNPNNAAALPGK	58	72	0.0054	14.0
	DQB1*03:03	VPGVNPNNAAALPGK	58	72	0.21	1.1
	DQB1*05:01	VPGVNPNNAAALPGK	58	72	0.0073	13.0
	DQB1*05:03	NVNNLARTTPDRQAA	33	47	0.05	4.1

Summary of predicted epitopes by Immune Epitope Database (IEDB) prediction tool, including information about matched alleles, epitope sequence, the corresponding start and end amino acid positions and binding affinity parameters (score and percentile rank). The epitopes shown correspond to those with the highest score matched with alleles significantly different across allergy cohort with population frequency greater than 2%. ^1^ Score: >0.5 high binding affinity (green); 0.2–0.5 moderate binding affinity (yellow); <0.2 low binding affinity (orange). ^2^ Percentile rank: <2 strong binders (green); 2–10 weak binders (yellow); >10 not an expected epitope (orange).

## Data Availability

The data presented in this study are available on request from the corresponding author due to ethical reasons.

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
