# Peer review of "Association Between Peach and Olive Pollen Non-Specific Lipid Transfer Protein Allergy and HLA Class II Phenotype"

_ijms, 2025, doi:10.3390/ijms26167755_

Round 1

Reviewer 1 Report

Comments and Suggestions for Authors

Introduction section is very well structured and complete, but I suggest authors to include a more detailed paragraph about HLA genetic and protein structure. HLA readers may not need this, but I think this is a paper more focused in the “allergy” field so maybe it will help the understanding of allergy researchers and others.

Lines 96-103. It seems to be more accurate that these lines were placed at the beginning of Material and Methods section. Patients stratification is not a result of the study.

Tables 2 and 3. Authors state in M&M section that HLA high-resolution typing has been carried out for the study. However, only DRB1*04, DRB1*11, DQB1*03 and DQB1*05 groups of alleles are shown in high resolution. I presume that this is because other DRB1 and DQB1 two-digits alleles show very low frequencies and no significative results are found with them, but the omission of the other alleles should be explained in M&M or Results section in this way.

Author Response

We appreciate the timely indications regarding this study that have helped improve the scientific quality of this work and its correct presentation.

Comments 1: Introduction section is very well structured and complete, but I suggest authors to include a more detailed paragraph about HLA genetic and protein structure. HLA readers may not need this, but I think this is a paper more focused in the “allergy” field so maybe it will help the understanding of allergy researchers and others.

Response 1: Following the indication of the reviewer, an extended explanation has been added in lines 71 to 81 with the requested information about HLA class II molecules.

Comments 2: Lines 96-103. It seems to be more accurate that these lines were placed at the beginning of Material and Methods section. Patients stratification is not a result of the study.

Response 2: We agree with the reviewer's indication. Patients stratification has been replaced at lines 448 to 454, in section 4. Materials and Methods.

Comments 3: Tables 2 and 3. Authors state in M&M section that HLA high-resolution typing has been carried out for the study. However, only DRB1*04, DRB1*11, DQB1*03 and DQB1*05 groups of alleles are shown in high resolution. I presume that this is because other DRB1 and DQB1 two-digits alleles show very low frequencies and no significative results are found with them, but the omission of the other alleles should be explained in M&M or Results section in this way.

Response 3: As the reviewer indicates, high resolution genotype was performed for all alleles, but we only showed those whose frequency was significantly different between groups or compared with the control group. Clarifying notes have been included in 2.4 section of Results (lines 158 to 160), and also in Tables 2 and 3 footnote.

Reviewer 2 Report

Comments and Suggestions for Authors

This manuscript presents a relevant and well-designed study on the association between HLA class II alleles and sensitization to Ole e 7 and Pru p 3. The topic is of interest and the findings contribute valuable insights. However, some revisions are needed to improve clarity, strengthen the analysis, and enhance the overall presentation.

  1. While the use of the IEDB platform for T-cell epitope prediction is methodologically sound, the study lacks experimental validation of the predicted epitopes. Functional assays such as ELISPOT, T-cell proliferation, or HLA-peptide tetramer staining are not included to confirm T-cell recognition. The authors are encouraged to either include functional validation or clearly discuss this limitation and its implications in the Discussion section.
  2. The manuscript uses a threshold of 0.35 kUA/L to define sensitization in Section 2.1 (lines ~97–103), which is a common cutoff. However, this threshold alone may not adequately distinguish between clinically relevant sensitization and asymptomatic sensitization. The authors are encouraged to provide a more detailed justification for relying solely on this cutoff without incorporating clinical challenge tests or symptom assessments. Additionally, a discussion on how the chosen threshold might affect the interpretation of the results would strengthen the manuscript.

Author Response

We appreciate the timely indications regarding this study that have helped improve the scientific quality of this work and its correct presentation.

Comments 1: This manuscript presents a relevant and well-designed study on the association between HLA class II alleles and sensitization to Ole e 7 and Pru p 3. The topic is of interest and the findings contribute valuable insights. However, some revisions are needed to improve clarity, strengthen the analysis, and enhance the overall presentation.

While the use of the IEDB platform for T-cell epitope prediction is methodologically sound, the study lacks experimental validation of the predicted epitopes. Functional assays such as ELISPOT, T-cell proliferation, or HLA-peptide tetramer staining are not included to confirm T-cell recognition. The authors are encouraged to either include functional validation or clearly discuss this limitation and its implications in the Discussion section.

Response 1: Indeed, the experimental validation suggested by the reviewer is extremely interesting and useful and should be addressed in future confirmatory studies, as noted in the limitations paragraph. Authors have described different experiments for this purpose, as we have mentioned throughout the discussion and, following the reviewer indication, we have added an additional paragraph explaining this limitation (lines 374 to 382).

Comments 2: The manuscript uses a threshold of 0.35 kUA/L to define sensitization in Section 2.1 (lines ~97–103), which is a common cutoff. However, this threshold alone may not adequately distinguish between clinically relevant sensitization and asymptomatic sensitization. The authors are encouraged to provide a more detailed justification for relying solely on this cutoff without incorporating clinical challenge tests or symptom assessments. Additionally, a discussion on how the chosen threshold might affect the interpretation of the results would strengthen the manuscript.

As indicated by the reviewer, the cut-off described in this work was the one recommended by the manufacturer, but this limitation was considered in the selection of patients. On the one hand, as it can be seen in Table 1 and Figure 3A, mean levels of sIgE for Ole e 7 and Pru p 3 were considerably higher than 0.35 kUA/L. To emphasise this, the following explanation has been added to section 2.6 (lines 276 to 279): “Although the positivity threshold recommended by the manufacturer to consider a patient sensitised to an allergen was 0.35 kUA/L, most of the samples tested had sIgE levels to Ole e 7 and to Pru p 3 considerably higher than the mentioned threshold”. On the other hand, a previous study performed by our group defined a recalculated cut-off for sIgE to Ole e 7 of 0.08 kUA/L, based on relevant association with allergic symptomatology (DOI: 10.3389/falgy.2023.1241650). Hence, the standard cut-off of 0.35 kUA/L was convenient for the stratification - 70.29% sensitivity and 81.95% specificity. In the case of Pru p 3, no similar study exists, although 88.6% of the determinations of patients sensitised to Pru p 3 were above 3.5 kUA/L, 10 times over the established threshold.

Likewise, we agree with the reviewer, that the sentence “A group of 62 allergic patients were recruited and stratified into three groups according to specific IgE (sIgE) levels, irrespective of clinical presentation” is confusing. Therefore, it has been rectified in line 448. As detailed in sections 2.1, 2.2, and 2.3, the clinical data supported the diagnosis beyond sIgE levels. In fact, all patients were selected after attending Allergy Department with allergic symptomatology, and a clinical diagnosis was established prior to their inclusion in the study.